# Regional differences in primary healthcare utilization in Java Region—Indonesia

**Ratna Dwi Wulandari**[1,2]*, **Agung Dwi Laksono**[2,3], **Nikmatur Rohmah**[4], **Hadi Ashar**[3]

**1** Faculty of Public Health, Universitas Airlangga, Surabaya, Indonesia, **2** The Airlangga Centre for Health Policy (ACeHAP), Surabaya, Indonesia, **3** National Research and Innovation Agency Republic of Indonesia, Jakarta, Indonesia, **4** Faculty of Health Science, Muhammadiyah University of Jember, East Java, Indonesia

\* ratna-d-w@fkm.unair.ac.id

## Abstract

### Background

Policymakers must understand primary healthcare utilization disparity to minimize the gap because they must seek fair service for every citizen. The study analyzes regional differences in primary healthcare utilization in Java Region-Indonesia.

### Methods

The cross-sectional research analyzes secondary data from the 2018 Indonesian Basic Health Survey. The study setting represented Java Region-Indonesia, and the participants were adults 15 years or more. The survey explores 629,370 respondents. The study used primary healthcare utilization as an outcome variable and province as the exposure variable. Moreover, the study employed eight control variables (residence, age, gender, education, marital, employment, wealth, and insurance). The study evaluated data using binary logistic regression in the final step.

### Results

People in Jakarta are 1.472 times more likely to utilize primary healthcare than those in Banten (AOR 1.472; 95% CI 1.332–1.627). People in Yogyakarta are 1.267 times more likely to use primary healthcare than those in Banten (AOR 1.267; 95% CI 1.112–1.444). In addition, people in East Java are 15% less likely to utilize primary healthcare than those in Banten (AOR 0.851; 95% CI 0.783–0.924). Meanwhile, direct healthcare utilization was the same between West Java, Central Java, and Banten Province. They are sequentially starting from the minor primary healthcare utilization: East Java, Central Java, Banten, West Java, Yogyakarta, and Jakarta.

### Conclusion

Disparities between regions exist in the Java Region-Indonesia. They are sequentially starting from the minor primary healthcare utilization: East Java, Central Java, Banten, West Java, Yogyakarta, and Jakarta.

**Data Availability Statement:** The data supporting this study's findings are available from the National Institute of Health Research and Development of Indonesia's Ministry of Health. Still, restrictions apply to the availability of these data used under

license for the current study, so they are not publicly available. However, data are available from the authors upon reasonable request and with permission of the National Institute of Health Research and Development of Indonesia Ministry of Health via the web page http://www.litbang.kemkes.go.id/jasa-permintaan-data-riset.

**Funding:** The author(s) received no specific funding for this work.

**Competing interests:** The authors have declared that no competing interests exist.

## Introduction

According to the World Health Organization, primary healthcare is essential healthcare based on methods and techniques that are practical, scientific, socially acceptable, and universally accessible to individuals, families, and communities [1]. Puskesmas (Pusat Kesehatan Masyarakat/Center of Public Health) is a primary healthcare facility unique to Indonesia. The primary entry point to the government-run healthcare system is Puskesmas. In addition to general practice and primary clinics, Puskesmas is Indonesia's entry point to critical services. Every subdistrict, or 30,000 residents in Indonesia, must have access to primary healthcare [2]. At least 9,993 Puskesmas in Indonesia will serve around 250 million people through 2018 [3].

In Indonesia, Puskesmas, as the embodiment of primary healthcare, is the first-level health service facility in the era of national health insurance (NHI). Puskesmas are required to provide comprehensive health services, covering 144 diseases for participants. Services must have multiple benefits, both medical and non-medical; namely, the services provided are complete, ranging from preventive, promotive, curative, and rehabilitative. Promotive is a process for the community's ability to maintain and improve their health; One of the efforts made by the Puskesmas is health counseling. Meanwhile, prevention is an effort to take various actions to avoid health problems threatening ourselves and others in the future [4].

On the other hand, curative is a health effort to prevent the disease from getting worse through treatment; The Puskesmas provides the most appropriate and quickest possible treatment for each type of disease so that complete and immediate healing is achieved. Puskesmas serve this effort by providing medical personnel. Moreover, curative is an action or a series of activities aimed at ex-sufferers (patients who are no longer suffering from the disease), so they can generally interact in the social environment [4].

There are often some misconceptions about primary healthcare, namely the assumption that primary healthcare provides only basic care when primary healthcare provides essential care that can meet most of a person's health needs throughout life. Primary healthcare serves health for all ages for all diseases, including promotive, preventive, curative, rehabilitation, and palliative [1]. Several factors are related to the utilization of primary healthcare, such as the variety of services, the consultation process, affordability, and convenience. The situation correlates with patient comfort and satisfaction [5]. Patient satisfaction is the most crucial and is one of the main reasons patients visit primary healthcare [6]. Experience in Canada, several factors influence the use of primary healthcare. It includes doctor's practice, various services, incentives, combination payment schemes, electronic medical records, and efforts to improve quality, showing that primary healthcare transformation is achieved pluralistically [7].

Health development objectives are to increase the reach and distribution of quality health services for the community. With Indonesia's existing geographical conditions, it is challenging for the government to solve. So far, comprehensive health services have been centered in Java; some of the best-advanced healthcare facilities (hospitals) are in Java [8]. The Java region is a benchmark for success for other areas in Indonesia, including models for policies addressing gaps in primary healthcare utilization.

Achieving the primary healthcare target requires additional investment in more comprehensive health services. At the United Nations high-level universal health coverage meeting in 2019, countries committed to strengthening primary healthcare. The WHO recommends that each country allocate an additional budget of 1% of GDP [9]. All stakeholders, from the government and doctors to community members, need to be aware of primary healthcare's role and benefits [1]. Policy policymakers should pursue patient satisfaction in direct healthcare services by improving the quality of medical services, especially in rural areas [5]. According to the research background, the study analyzes regional differences in primary healthcare utilization in Java Region in Indonesia.

## Methods

### Study design and data source

We designed a cross-sectional study using secondary data from the 2018 Indonesian Basic Health Survey. The 2018 Indonesian Basic Health Survey was a national-scale cross-sectional survey conducted by the Ministry of Health of the Republic of Indonesia. The survey collected data from May to July 2018 through Household and Individual Instruments interviews.

The survey included all Indonesian households. This survey employs the sample framework of the 2018 National Socio-Economic Survey (conducted by the Central Statistics Agency in March 2018). Furthermore, in the 2018 Socio-Economic Survey, the 2018 Indonesian Basic Health Survey visited a target sample of 300,000 households from 30,000 census blocks [10].

The PPS (probability proportional to size) method is used in the 2018 Indonesian Basic Health Survey, with systematic linear sampling in two stages: Stage 1: Implicit stratification based on the welfare strata of all census blocks from the 2010 Population Census. PPS selected the sample survey as the sampling frame from a master frame of 720,000 census blocks from the 2010 Population Census, of which 180,000 were chosen (25%). The survey used the PPS method to determine several census blocks in each urban/rural strata per regency/city to create a Census Block Sample List. A total of 30,000 Census Blocks have been selected. Stage 2: Using systematic sampling, choose ten households in each Census Block with the highest implicit education stratification completed by the Head of Household to maintain the representation of the diversity value of household characteristics. As part of the 2018 Indonesian Basic Health Survey, all selected household members will be interviewed [10].

The study population comprised all adults (15 years old) in Indonesia's Java Region. According to the sampling methods, we obtained a weighted sample of 213,140 adults for this study.

### Outcome variable

The study's outcome variable was primary healthcare utilization—adults' access to primary healthcare; whatever the reason for the need for such access, whether outpatient or inpatient, was the utilization of primary healthcare. Outpatient care was limited to the previous month, while inpatient care was limited to last year. The survey asked respondents to recall the correct outpatient and inpatient incidences [10]. Primary healthcare utilization comprises unutilized and utilized.

### Exposure variable

We employed region or province as an exposure variable in this study. The region comprises all provinces (six) on Java Island, namely Jakarta, West Java, Central Java, Yogyakarta, East Java, and Banten.

### Control variables

Moreover, the study employed eight factors as control variables. The eight factors were the type of residence, age group, gender, education level, marital status, employment status, wealth status, and health insurance ownership. The study divided the available residence into two categories: urban and rural. We used the Indonesian Central Statistics Agency's provisions for urban-rural categorization in the survey. Furthermore, the study calculated age based on the most recent birthday, and the age group comprises $\leq 17$, 18–64, and $\geq 65$. Gender includes two categories: male and female.

The study also classified marital status into three categories: single, married/living with a partner, and divorced/widowed. The study defined education as accepting the respondent's most recent diploma. The study has four levels of education: no education, primary, secondary, and higher education. Meanwhile, there are two employment options: unemployed and employed.

The pool used the wealth index formula to determine wealth status in the study. The survey calculated the wealth index using a weighted average of a family's total spending. Meanwhile, the survey calculated the wealth index using primary household expenditures such as health insurance, food, and lodging, among other things. In addition, the poll classified income into five categories: poorest, poorer, middle, wealthier, and most prosperous [11, 12]. According to the survey, there are two types of health insurance owners: uninsured and insured.

## Study setting

We are setting the study to represent the Java Region in Indonesia.

## Data analysis

The Chi-Square test was used in the early stages of the sample to produce a bivariate comparison between region (province) with other variables. Furthermore, we used a collinearity test to ensure that the independent variables in the final regression model did not have a strong relationship. The study's final point employed binary logistic regression (enter method). The survey used the last test to examine the multivariate relationship between all independent variables and primary healthcare utilization. We declare it's statistically significant at $p < 0.050$. With 95% confidence intervals, adjusted odds ratios (AOR) were presented (95% CI). Throughout the statistical analysis process, we employed the IBM SPSS 26 application.

## Ethical approval

The 2018 Indonesian Basic Health Survey received Ethical Clearance from the National Ethics Committee (LB.02.01/2/KE.024/2018). During data collection, the survey used informed consent, which accounted for aspects of the data collection procedure, such as voluntariness and confidentiality. Respondents provided written permission.

## Results

The analysis found that the average primary healthcare utilization in Java Region in 2018 was 5.1%. Meanwhile, Table 1 shows descriptive statistics of direct healthcare utilization and respondents' characteristics in Indonesia in 2018. The result indicates that Yogyakarta Province has the highest primary healthcare utilization compared to other provinces in the Java Region. Based on the type of residence, a minor proportion of urban areas is in Central Java Province. According to the age group, 18–64 dominated all provinces. Moreover, regarding gender, males ruled in West Java and Banten Provinces.

Based on marital status, Table 1 indicates marriage dominated all regions. Regarding education level, primary education leads all provinces, except Jakarta Provinces, which are led by secondary education. According to employment status, employed dominated all areas. The most prosperous lead in Jakarta, West Java, Yogyakarta, and Banten Province is based on wealth status. Regarding health insurance ownership, insured people dominated all regions.

Table 2 shows the binary logistic regression of primary healthcare utilization in Java Region-Indonesia. We used "unutilized primary healthcare" as a reference. Based on the province, people in Jakarta are 1.472 times more likely to utilize primary healthcare than those in

**Table 1. Descriptive statistic of primary healthcare utilization and respondents' characteristics in Java Region in Indonesia, 2018 (n = 213,140).**

| Demographic Characteristics | Province | | | | | | p-value |
|---|---|---|---|---|---|---|---|
| | Jakarta (n = 9,711) | West Java (n = 48,491) | Central Java (n = 63,118) | Yogyakarta (n = 8,312) | East java (n = 68,747) | Banten (n = 14,761) | |
| **Primary Healthcare Utilization** | | | | | | | < 0.001 |
| Unutilized | 93.8% | 94.7% | 94.8% | 93.6% | 95.4% | 95.4% | |
| Utilized | 6.2% | 5.3% | 5.2% | 6.4% | 4.6% | 4.6% | |
| **Type of residence** | | | | | | | < 0.001 |
| Urban | 100.0% | 74.1% | 51.3% | 72.9% | 52.3% | 71.9% | |
| Rural | - | 25.9% | 48.7% | 27.1% | 47.7% | 28.1% | |
| **Age Group** | | | | | | | < 0.001 |
| ≤ 17 | 5.7% | 7.5% | 6.8% | 5.3% | 6.2% | 7.0% | |
| 18–64 | 88.6% | 84.8% | 82.0% | 82.7% | 83.5% | 88.2% | |
| ≥ 65 | 5.7% | 7.7% | 11.2% | 12.1% | 10.3% | 4.8% | |
| **Gender** | | | | | | | < 0.001 |
| Male | 49.7% | 50.4% | 49.0% | 48.7% | 48.8% | 50.8% | |
| Female | 50.3% | 49.6% | 51.0% | 51.3% | 51.2% | 49.2% | |
| **Marital status** | | | | | | | < 0.001 |
| Never in union | 25.5% | 22.5% | 21.4% | 23.3% | 19.6% | 24.4% | |
| Married/Living with a partner | 66.9% | 69.4% | 69.2% | 66.9% | 69.7% | 68.6% | |
| Divorced/Widowed | 7.5% | 8.1% | 9.5% | 9.9% | 10.7% | 7.0% | |
| **Education level** | | | | | | | < 0.001 |
| No education | 2.6% | 3.5% | 7.6% | 6.0% | 8.4% | 5.1% | |
| Primary | 38.9% | 62.5% | 63.4% | 41.6% | 59.3% | 53.9% | |
| Secondary | 44.0% | 26.7% | 22.5% | 38.9% | 25.4% | 32.5% | |
| Higher | 14.5% | 7.3% | 6.5% | 13.5% | 7.0% | 8.5% | |
| **Employment status** | | | | | | | < 0.001 |
| Unemployed | 39.7% | 42.9% | 32.7% | 32.9% | 34.4% | 43.5% | |
| Employed | 60.3% | 57.1% | 67.3% | 67.1% | 65.6% | 56.5% | |
| **Wealth status** | | | | | | | < 0.001 |
| Poorest | 1.0% | 16.3% | 25.3% | 18.4% | 22.0% | 8.6% | |
| Poorer | 4.7% | 19.3% | 20.7% | 19.4% | 20.5% | 14.6% | |
| Middle | 16.3% | 18.1% | 18.7% | 15.4% | 18.2% | 19.5% | |
| Richer | 25.0% | 18.1% | 18.3% | 19.8% | 20.0% | 22.9% | |
| Richest | 53.0% | 28.2% | 17.1% | 27.0% | 19.3% | 34.4% | |
| **Health Insurance** | | | | | | | < 0.001 |
| Uninsured | 15.5% | 34.4% | 31.2% | 17.5% | 37.4% | 35.3% | |
| Insured | 84.5% | 65.6% | 68.8% | 82.5% | 62.6% | 64.7% | |

Banten (AOR 1.472; 95% CI 1.332–1.627). People in Yogyakarta are 1.267 times more likely to use primary healthcare than those in Banten (AOR 1.267; 95% CI 1.112–1.444). In addition, people in East Java are 15% less likely to utilize primary healthcare than those in Banten (AOR

**Table 2. The result of binary logistic regression of primary healthcare utilization in Java Region in Indonesia, 2018 (n = 213,140).**

| Predictor | Utilized Primary Healthcare | | | |
|---|---|---|---|---|
| | p-value | AOR | 95% CI | |
| | | | Lower Bound | Upper Bound |
| Province: Jakarta | *<0.001 | 1.472 | 1.332 | 1.627 |
| Province: West Java | 0.492 | 1.028 | 0.950 | 1.114 |
| Province: Central Java | 0.123 | 0.936 | 0.861 | 1.018 |
| Province: Yogyakarta | *<0.001 | 1.267 | 1.112 | 1.444 |
| Province: East Java | *<0.001 | 0.851 | 0.783 | 0.924 |
| Province: Banten | - | - | - | - |
| Residence: Urban | **0.001 | 1.080 | 1.034 | 1.129 |
| Residence: Rural | - | - | - | - |
| Age: ≤ 17 | - | - | - | - |
| Age: 18–64 | 0.292 | 1.062 | 0.950 | 1.187 |
| Age: ≥ 65 | *<0.001 | 1.522 | 1.342 | 1.726 |
| Gender: Male | - | - | - | - |
| Gender: Female | *<0.001 | 1.326 | 1.267 | 1.387 |
| Marital: Never in union | - | - | - | - |
| Marital: Married/Living with partner | *<0.001 | 1.716 | 1.592 | 1.849 |
| Marital: Divorced/Widowed | *<0.001 | 2.174 | 1.977 | 2.391 |
| Education: No Education | - | - | - | - |
| Education: Primary | 0.102 | 0.942 | 0.876 | 1.012 |
| Education: Secondary | *<0.001 | 0.560 | 0.513 | 0.611 |
| Education: Higher | *<0.001 | 0.302 | 0.262 | 0.348 |
| Employment: Unemployed | - | - | - | - |
| Employment: Employed | *<0.001 | 0.805 | 0.769 | 0.842 |
| Wealth: Poorest | - | - | - | - |
| Wealth: Poorer | 0.252 | 0.966 | 0.910 | 1.025 |
| Wealth: Middle | 0.160 | 0.957 | 0.901 | 1.017 |
| Wealth: Richer | *<0.001 | 0.866 | 0.814 | 0.922 |
| Wealth: Richest | *<0.001 | 0.617 | 0.576 | 0.661 |
| Health insurance: Uninsured | - | - | - | - |
| Health insurance: Insured | *<0.001 | 1.841 | 1.757 | 1.929 |

Note: *$p<0.001$

**$p<0.010$

0.851; 95% CI 0.783–0.924). Meanwhile, there was no significant difference in direct healthcare utilization between West Java, Central Java, and Banten Province. They are sequentially starting from the minor primary healthcare utilization: East Java, Central Java, Banten, West Java, Yogyakarta, and Jakarta.

Moreover, the study also found eight control variables significantly correlated with primary healthcare utilization. Regarding the type of residence, people in urban areas are 1.080 times more likely to use primary healthcare than those in rural areas (AOR 1.080; 95% CI 1.034–1.129). According to the age group, those ≥ 65 are 1.522 times more likely than those ≤ 17 to utilize primary healthcare (AOR 1.522; 95% CI 1.342–1.726). Based on gender, females are 1.326 times more likely to use primary healthcare than males (AOR 1.326; 95% CI 1.267–1.387).

Married or living with a partner is 1.716 times more likely to use primary healthcare than never in a union (AOR 1.716; 95% CI 1.592–1.849). Furthermore, divorced or widowed are

2.174 times more likely to utilize primary healthcare than those never in a union (AOR 2.174; 95% CI 1.977–2.391). Based on education level, all education levels are less likely than no school to utilize primary healthcare, except primary education, which is not significantly related to no education.

According to employment, the employed are 20% times less likely to use primary healthcare than the unemployed (AOR 0.805; 95% CI 0.769–0.842). All wealth statuses are less likely than the poorest to use primary healthcare, except for the poorer and the middle. There are no significant differences among the poorest in utilizing primary healthcare. Based on health insurance ownership, the insured are 1.841 times more likely than the uninsured to use primary healthcare (AOR 1.841; 95% CI 1.757–1.929).

## Discussion

Essential health services are the first level of health services and are the first contact of the population with the healthcare system. The primary healthcare management policy aims to increase the utilization of essential health services in Indonesia. One of these policies is establishing the Regional Public Service Agency (RPSA). This RPSA makes primary healthcare more developed because the funds/financial support issued follow the needs and are not dependent on the health department [13]. The utilization of primary healthcare will be a critical factor in Indonesia's successful use of health infrastructure [14]. The Indonesian government has implemented a series of Universal Health Coverage (UHC) policy reforms that include the integration of the government insurance scheme into the National Health Insurance (NHI), expanding the provider network, and accreditation of all health facilities [15]. However, the study results show that UHC in Indonesia still faces the burden of state finances for healthcare and a lack of healthcare facilities, especially in primary health units [16].

The results show differences in primary healthcare utilization between regions in Java Region in Indonesia. The utilization of essential health services between areas in the Java Region of Indonesia is still not evenly distributed. The results are similar to the geographic inequalities in health services utilization in 497 districts in Indonesia. The study results indicate a significant variation in the utilization of health services at the district level and a minor variation at the provincial level [17, 18]. Studies on the reform of the Indonesian public health system also find indications that post-reform, the gap is widening in the propensity of the poor to file insurance claims depending on their level of access to health centers [19]. The study supports previous studies that reported disparities in the utilization of primary health among regions in rural Indonesia. Regions in western Indonesia tend to have better utilization of Puskesmas in rural areas [8, 20, 21].

The study indicates people in urban areas are more likely to use primary healthcare than those in rural areas. Rural people have more significant barriers to using primary healthcare than people in urban areas. The results support the study in the US, which states that compared to urban residents, rural residents have lower access to primary care providers [22, 23]. However, the survey on the satisfaction level of primary healthcare did not show a significant difference between rural and urban dwellings [24]. Studies in India also reveal rural-urban inequalities that do not benefit the rural population and the utilization of healthcare for the elderly [25]. This disparity also exists in China; a study reports that rural residents in China are still underutilizing healthcare when compared to their urban counterparts [26]. A similar study in Ghana shows a generally low utilization of health services among rural communities [27].

All age groups are more likely than ≤ 17 to utilize primary healthcare, and young adults and above mainly use primary healthcare. These results support an integrative review study

that states that age is the predisposing factor that most influences the use of direct health services by the elderly [28]. The survey in Gomoa confirmed that age was the key factor determining the utilization of health services in the sub-district [29]. Studies in the Chongqing or Guizhou areas also report that age is associated with knowledge, utilization of health education, and satisfaction in the primary healthcare sector. The situation is related to previous treatment experience and degenerative disease factors [30].

All marital statuses are more likely to use primary healthcare than those who are never in a union, and those in marital status have a greater need to utilize direct health services. In line with the results of this study, marital status determines the utilization of primary health services [31]. Studies in China also stated that marital status was associated with knowledge, utilization, and satisfaction with health education in primary healthcare [32, 33]. Compared to unmarried respondents, married respondents tend to make outpatient visits, and married patients use more healthcare services at the secondary and primary healthcare levels [34].

The study shows education level related to primary healthcare utilization in Java Region in Indonesia. Those not educated tend to use more essential health services than those with secondary and higher education. This study supports a study in Ghana, which reported that education had a positive and significant effect on health services utilization in rural communities [27]. The level of education also has a statistically significant relationship with the likelihood of seeking services and the level of services used in dealing with simple ailments in the Iranian Shiraz community [35]. Education level was also significantly associated with using primary healthcare for antenatal care [36]. This result is different from the study in Gomoa, which reported that education level was insignificant as a determinant of primary healthcare utilization [29].

According to employment, the employed are less likely to use primary healthcare than the unemployed. People who work generate income, and employed people may have a higher likelihood of using primary healthcare. Another study states that income level is significantly related to the probability of finding services and the services used in dealing with simple diseases [35]. This result is different from the study in Gomoa, which confirmed that the average monthly income was insignificant as a determinant of primary health service utilization [29].

All wealthy statuses were less likely to use primary healthcare than the poorest, except for the poorer and middle class. There is no significant difference between the poorest communities in direct health services, and people mostly use primary healthcare, with the most impoverished. The situation is in line with the results of a study in Ethiopia which stated that the supporting factor in the form of monthly income determines the utilization of primary health services. Low household and socio-economic status encourage direct health services [31]. Moreover, wealth status is also related to health insurance ownership, and wealthier people tend to have health insurance [37, 38].

The insured is more likely than the uninsured to use primary healthcare based on the possession of health insurance. People who have health insurance make more use of direct health services. This study's results align with reflections on the impact of subsidy policies on the utilization of various types of healthcare facilities in Indonesia. The results show that subsidized NHI premiums for the poor at the first level of healthcare facilities can increase the likelihood of outpatient visits to primary healthcare [39, 40]. The study in Shiraz also stated that insurance and type had a significant relationship with the probability of finding services and the level of service used in dealing with simple illnesses [35]. However, previous research in Indonesia found that general health insurance for the poor and public health insurance had minimal impact on the utility of health facilities as measured by outpatient and inpatient visits by program beneficiaries [41]. The NHI program increases the utilization of outpatient and inpatient care in the contribution group. To a lesser extent, those with subsidized insurance have

increased access to inpatient-only facilities [42]. Expanding health insurance coverage reduces sociodemographic disparities in Indonesia's access to maternal health services. However, significant differences in utilization remain across regions and economic subgroups [43].

The government should minimize inter-regional service disparities in its territory, including differences in health services. To overcome this situation, the government has made several efforts. Among them encourage the expansion of NHI membership by regulating in more detail assistance or premium subsidies for NHI membership for the poor and requiring employers to bear the NHI membership premium for their workers [37, 44]. On the other hand, the government is also expanding the Puskesmas network by establishing *Pondok Kesehatan Desa (Ponkesdes)* or Village Health Cottages for each village [45]. Furthermore, the government is trying to collaborate with more primary clinics and general practices in the cooperation mechanism within NHI [46].

## Strength and limitation

The study examines a large amount of data to represent information on a national scale. On the other hand, the study examines secondary data; thus, the accepted variables limit the factors analyzed. Several other factors related to primary healthcare utilization discovered in previous studies, such as travel time, travel cost, and disease type, cannot be studied [17, 47].

## Conclusion

Based on the study results, the study concluded that disparities between regions exist in the Java Region in Indonesia. The study listed East Java, Central Java, Banten, West Java, Yogyakarta, and Jakarta in order of minor primary healthcare utilization.

Interventions to increase primary healthcare utilization focusing on regions are insufficient as the primary strategy to address health disparities at the population level. Future policies should concentrate more on interventions targeting social determinants of upstream health, targeting this study's results. Efforts should focus on increasing interventions for those who live in rural areas, are younger, have never been in a union, are better educated, are employed, have better wealth status, and are uninsured.

## Acknowledgments

The author would like to thank the National Institute of Health Research and Development, which has agreed to allow the authors to analyze the data set in this article.

## Author Contributions

**Conceptualization:** Ratna Dwi Wulandari, Agung Dwi Laksono.

**Data curation:** Nikmatur Rohmah, Hadi Ashar.

**Formal analysis:** Ratna Dwi Wulandari, Agung Dwi Laksono.

**Funding acquisition:** Ratna Dwi Wulandari.

**Investigation:** Nikmatur Rohmah, Hadi Ashar.

**Methodology:** Ratna Dwi Wulandari, Agung Dwi Laksono.

**Project administration:** Hadi Ashar.

**Resources:** Ratna Dwi Wulandari.

**Software:** Agung Dwi Laksono, Nikmatur Rohmah.

**Validation:** Nikmatur Rohmah, Hadi Ashar.

**Visualization:** Nikmatur Rohmah, Hadi Ashar.

**Writing – original draft:** Agung Dwi Laksono, Nikmatur Rohmah, Hadi Ashar.

**Writing – review & editing:** Ratna Dwi Wulandari.

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
