## [Editor Report · Decision Letter 0]

6 Jan 2023

PONE-D-22-29798

Regional Differences in Primary Health Care Utilization in Java Region - Indonesia

PLOS ONE

Dear Dr. Wulandari,

Thank you for submitting your manuscript to PLOS ONE. After careful consideration, we feel that it has merit but does not fully meet PLOS ONE’s publication criteria as it currently stands. Therefore, we invite you to submit a revised version of the manuscript that addresses the points raised during the review process.

We look forward to receiving your revised manuscript.

Kind regards,

Retno Asti Werdhani, M.Epid, Ph.D

Academic Editor

PLOS ONE

 Journal Requirements:

2. We note that Figure 1 in your submission contain [map/satellite] images which may be copyrighted. All PLOS content is published under the Creative Commons Attribution License (CC BY 4.0), which means that the manuscript, images, and Supporting Information files will be freely available online, and any third party is permitted to access, download, copy, distribute, and use these materials in any way, even commercially, with proper attribution. For these reasons, we cannot publish previously copyrighted maps or satellite images created using proprietary data, such as Google software (Google Maps, Street View, and Earth). For more information, see our copyright guidelines: http://journals.plos.org/plosone/s/licenses-and-copyright.

Additional Editor Comments:

General comments:

1. Important issue to find the gap of health care utilization and may be factor related to health seeking behavior

2. Need further discussion to overcome the gap of health care utilization because there should not be any gap for health care utilization (health for all), example from another region/country/references so that it will in line with the recommendation mentioned in conclusion.

Abstract

1. Need more information on why policymakers need to understand primary health care utilization disparity to minimize the gap, therefore …

2. It’s not common for use interpret OR 0.851 as 0.851 times less likely. Usually, we say 1-0.851 = 15 % less likely

Background/Introduction

1. Please mention/explore more about the type/example of service provided for preventive, promotive, curative, and rehabilitative in Puskesmas. (5W1H Puskesmas) in background

2. The detail aim of the study hasn’t been mentioned yet. It’s important for the reader to see the aim of the study in the background so that they can see the redline between the aim and result.

3. Why only Java region, not the whole region in Indonesia

Method:

1. Define utilization: for disease cure seeking behavior, preventive seeking behavior?

2. For data analysis, it’s better to mention relationship between which variables using which statistical tools and when we declare it’s statistically significant or not? Or relationship?

3. Outcome variables, divided into what category? Was it utilized or not, or outpatient or inpatient

4. Justification why 25% selected sample from the population

5. How did you select 30.000 cencus blocks?

6. Table 2 presentation: please mention what method did you use in binary logistic regression for multivariate analysis? Enter/stepwise/forward/backward? Which group is reference line?

Result:

1. Interesting presentation using phc utilization map

2. Can the Colour of figure 1 not all purple? Can it change into more contrast colour like blue, red, yellow to divide the category?

3. “Based on marital status, married dominated all regions. Regarding education level, primary education leads all provinces, except Jakarta Provinces, which are led by secondary education. According to employment status, employed dominated all areas. The most prosperous lead in Jakarta, West Java, Yogyakarta, and Banten Province is based on wealth status. Regarding health insurance ownership, insured people dominated all regions.” � please write the reference table

4. 0.851 times less likely. It’s not common for use interpret OR 0.851 as 0.851 times less likely. Usually, we say 1-0.851 = 15 % less likely…

5. 0.805 times less likely. Same notes with the above feedback

Discussion:

1. Why east java less likely and Jakarta and Yogyakarta are more likely? � east java healthier people, better health system?

2. “Regions in western Indonesia tend to have better utilization of Puskesmas in rural areas [8,20]. � can you explore more on this? The example/comparison between western and rural areas? Is there no rural area in western? Because you compare between western and rural area

3. “Rural people have more significant barriers to using primary health care than people in urban areas” � please explore more what do you mean by significant barriers?

4. “All age groups are more likely than ≤ 17 to utilize primary health care, and young adults and above mainly use primary health care” � is it typo? < 17 is part of age groups?

5. “Studies in the Chongqing or Guizhou areas also report that age is associated with knowledge, utilization of health education, and satisfaction in the primary health care sector” � can you explore more about this? Why age is associated with those factors? Was it supporting or obstacles?

6. “the employed are less likely to use primary health care than the unemployed” � author discussed about the income.” � How about geographical access?

In conclusion:

“ Efforts should focus on increasing interventions for those who live in rural areas, are younger, have never been in a union, are better educated, are employed, have better wealth status, and are uninsured “ � it would be better if there is an example of how to do it?

---

## [Author Response · Author response to Decision Letter 0]

6 Feb 2023

RESPONSES

Dear Dr. Wulandari,

We've checked your submission and before we can proceed, we need you to address the following issues:

1. We note that Figure 1 in your submission contain [map/satellite] images which may be copyrighted. All PLOS content is published under the Creative Commons Attribution License (CC BY 4.0), which means that the manuscript, images, and Supporting Information files will be freely available online, and any third party is permitted to access, download, copy, distribute, and use these materials in any way, even commercially, with proper attribution. For these reasons, we cannot publish previously copyrighted maps or satellite images created using proprietary data, such as Google software (Google Maps, Street View, and Earth). For more information, see our copyright guidelines: http://journals.plos.org/plosone/s/licenses-and-copyright.

Response: we decided to deletes the figure in the manuscript.

Additional Editor Comments:

General comments:

1. Important issue to find the gap of health care utilization and may be factor related to health seeking behavior

2. Need further discussion to overcome the gap of health care utilization because there should not be any gap for health care utilization (health for all), example from another region/country/references so that it will in line with the recommendation mentioned in conclusion.

Response: the narration suggested was added to the discussion. 

Abstract

1. Need more information on why policymakers need to understand primary health care utilization disparity to minimize the gap, therefore …

2. It’s not common for use interpret OR 0.851 as 0.851 times less likely. Usually, we say 1-0.851 = 15 % less likely

Response: the abstract was revised as suggested.

Background/Introduction

1. Please mention/explore more about the type/example of service provided for preventive, promotive, curative, and rehabilitative in Puskesmas. (5W1H Puskesmas) in background

Response: the narration was added in the introduction as suggested.

2. The detail aim of the study hasn’t been mentioned yet. It’s important for the reader to see the aim of the study in the background so that they can see the redline between the aim and result.

Response: The purpose of the study is at the end of the introduction.

According to the research background, the study analyzes regional differences in primary health care utilization in Java Region in Indonesia.

3. Why only Java region, not the whole region in Indonesia

Response: The Java region is a benchmark for success for other areas in Indonesia, including models for policies addressing gaps in primary healthcare utilization.

Method:

1. Define utilization: for disease cure seeking behavior, preventive seeking behavior?

Response: We define utilization as access to primary healthcare; whatever the reason for the need for such access.

2. For data analysis, it’s better to mention relationship between which variables using which statistical tools and when we declare it’s statistically significant or not? Or relationship?

Response: the data analysis was revised as suggested.

3. Outcome variables, divided into what category? Was it utilized or not, or outpatient or inpatient

Response: Primary healthcare utilization comprises unutilized and utilized. 

4. Justification why 25% selected sample from the population

Response: This amount is to meet the minimum number of samples that must be met.

5. How did you select 30.000 cencus blocks?

Response: The survey chose the method of probability proportional to size systematically.

6. Table 2 presentation: please mention what method did you use in binary logistic regression for multivariate analysis? Enter/stepwise/forward/backward? Which group is reference line?

Response: The binary logistic regression used enter method. We used “unutilized primary healthcare” as a reference.

Result:

1. Interesting presentation using phc utilization map

Response: Thank you

2. Can the Colour of figure 1 not all purple? Can it change into more contrast colour like blue, red, yellow to divide the category?

Response: The author decided to delete the map due to copyright issue.

3. “Based on marital status, married dominated all regions. Regarding education level, primary education leads all provinces, except Jakarta Provinces, which are led by secondary education. According to employment status, employed dominated all areas. The most prosperous lead in Jakarta, West Java, Yogyakarta, and Banten Province is based on wealth status. Regarding health insurance ownership, insured people dominated all regions.” � please write the reference table

Response: The reference table was added as suggested.

4. 0.851 times less likely. It’s not common for use interpret OR 0.851 as 0.851 times less likely. Usually, we say 1-0.851 = 15 % less likely…

Response: The results were revised as suggested.

5. 0.805 times less likely. Same notes with the above feedback

Response: The results were revised as suggested.

Discussion:

1. Why east java less likely and Jakarta and Yogyakarta are more likely? � east java healthier people, better health system?

Response: The author cannot further analyze health status or health system. This study uses secondary data, so the analysis is limited to the data that has been received.

2. “Regions in western Indonesia tend to have better utilization of Puskesmas in rural areas [8,20]. � can you explore more on this? The example/comparison between western and rural areas? Is there no rural area in western? Because you compare between western and rural area

Response: The cited studies limit the analysis to only rural areas in Indonesia, and the results show that western has better utilization of Puskesmas.

3. “Rural people have more significant barriers to using primary health care than people in urban areas” � please explore more what do you mean by significant barriers.

Response: Those who live in rural areas have slightly lower utilization of the Puskesmas than those in urban areas.

4. “All age groups are more likely than ≤ 17 to utilize primary health care, and young adults and above mainly use primary health care” � is it typo? < 17 is part of age groups?

Response: The ≤ 17 is the reference line in the age group. In the sentence, “all age groups” means all groups except the ≤ 17.

5. “Studies in the Chongqing or Guizhou areas also report that age is associated with knowledge, utilization of health education, and satisfaction in the primary health care sector” � can you explore more about this? Why age is associated with those factors? Was it supporting or obstacles?

Response: The narration was added as suggested.

The situation is related to previous treatment experience and degenerative disease factors.

6. “the employed are less likely to use primary health care than the unemployed” � author discussed about the income.” � How about geographical access?

Response:

In conclusion:

“ Efforts should focus on increasing interventions for those who live in rural areas, are younger, have never been in a union, are better educated, are employed, have better wealth status, and are uninsured “ � it would be better if there is an example of how to do it?

Response: The study limits recommendations by only providing the characteristics of policy targets according to the study's results. Further studies are needed to determine the right policy by involving stakeholders interested in this situation.

---

## [Editor Report · Decision Letter 1]

14 Mar 2023

Regional Differences in Primary Health Care Utilization in Java Region - Indonesia

PONE-D-22-29798R1

Dear Dr. Wulandari,

We’re pleased to inform you that your manuscript has been judged scientifically suitable for publication and will be formally accepted for publication once it meets all outstanding technical requirements.

Kind regards,

Retno Asti Werdhani, M.Epid, Ph.D

Academic Editor

PLOS ONE
---

## [Editor Report · Acceptance letter]

20 Mar 2023

PONE-D-22-29798R1 

Regional Differences in Primary Healthcare Utilization
in Java Region - Indonesia 

Dear Dr. Wulandari:

I'm pleased to inform you that your manuscript has been deemed suitable for publication in PLOS ONE. Congratulations! Your manuscript is now with our production department. 

Kind regards, 

on behalf of

Dr. Retno Asti Werdhani 

Academic Editor

PLOS ONE